# Application of Antimicrobial Rubber-Coated Cotton Gloves for Mangosteen-Peel-Extract-Mediated Biosynthesis of Ag–ZnO Nanocomposites

**DOI:** 10.3390/polym17010032

**Published:** 2024-12-26

**Authors:** Montri Luengchavanon, Ekasit Anancharoenwong, Sutida Marthosa, Theerakamol Pengsakul, Jidapa Szekely

**Affiliations:** 1Sustainable Energy Management Program, Wind Energy and Energy Storage Systems Centre (WEESYC), Faculty of Environmental Management, Prince of Songkla University, Hatyai 90110, Thailand; 2Centre of Excellence in Metal and Materials Engineering (CEMME), Engineering Faculty, Prince of Songkla University, Hatyai 90110, Thailand; 3Materials and Renewable Energy Research Group, Faculty of Science and Industrial Technology, Prince of Songkla University, Surat Thani Campus, Surat Thani 84000, Thailand; ekasit.a@psu.ac.th; 4Centre of Excellence in Membrane Science and Technology, Faculty of Science and Industrial Technology, Prince of Songkla University, Surat Thani Campus, Surat Thani 84000, Thailand; sutida.m@psu.ac.th; 5Health and Environmental Research Centre, Faculty of Environmental Management, Prince of Songkla University, Hatyai 90110, Thailand; theerakamol.p@psu.ac.th; 6Faculty of Medical Technology, Prince of Songkla University, Hatyai 90110, Thailand; jidapa.sz@psu.ac.th

**Keywords:** mangosteen peel, Ag nanoparticles, Zn nanoparticles, plant-extract-mediated biosynthesis, rubber gloves, antimicrobial activity

## Abstract

Nanocomposites based on metal nanoparticles (MNP) prepared with mangosteen (_mgt_) peel extract-mediated biosynthesis of Ag_mgt_/Zn_mgt_ have attracted considerable interest due to their potential for various practical applications. In this study, their role in developing antibacterial protection for rubber cotton gloves is investigated. The process of mangosteen-peel-extract-mediated biosynthesis produced Ag_mgt_/Zn_mgt_ nanocomposites with respective diameters of 23.84 ± 4.08 nm and 30.99 ± 5.73 nm, which were assessed in the context of antimicrobial rubber-coated gloves. The rubber glover surface exhibited a very dense deposition of the Ag+Zn_mgt_ nanocomposite, which subsequently demonstrated level 4 resistance to punctures under the ANSI-ISEA 105-2016 standard. This could be attributed to the Zn-cellulose double formation on the rubber surface. Notably, on testing the inhibition of bacterial growth, the extract with the Ag_mgt_ nanoparticles presented the least concentration capable of growth inhibition in comparison to the extracts with Zn_mgt_ and Ag+Zn_mgt_ nanoparticles. Each of the mangosteen extracts was shown to inhibit bacterial growth when tested against both Gram-positive cocci and Gram-negative bacilli, with MIC in the range 40–320 µg/mL. The growth of drug-resistant bacteria (MRSA) could also be inhibited with an MIC value of 160 µg/mL, and with 30 min of contact, gloves with respective coatings of Zn_mgt_ and Ag+Zn_mgt_ extract nanocomposites were shown to inhibit *K. pneumoniae* and MRSA. However, while effective bacterial inhibition occurred with the suspensions, the coatings on glove surfaces required a lengthy incubation period (contact time) of at least 30 min for efficacy.

## 1. Introduction

Mangosteen fruits in Thailand have mainly been produced for exports, which benefit the farmers financially. Mangosteens are among the major fruits generating profits when exported from Thailand to many other countries, such as Japan, Taiwan, and China. Currently, there is a big problem with an imbalance of supply and demand [1]. The quantity of mangosteens that cannot be exported to other countries should instead be consumed domestically, when there is over supply. The mangosteen peel waste from restaurants, famers, and markets has been increasing, and there is a need for treating this waste to create added value.

In rubber glove production, estimating an appropriated curing degree involves the consideration of many mechanical properties fundamental to rubber gloves. Generally, rubber gloves have been tested for conventional measures of trust in experiments for protection, to determine time and temperature in the rubber-curing process to meet the required properties [2]. The typical physical resistance tests are conducted to assess abrasion, cut, and punch as well as tear resistances. The rubber gloves use natural rubber (NR) that can be combined with metal nanoparticles (MNPs) for added value in various applications. The linear polymer chains of NR must be cross linked with covalent bonds, forming a vulcanised molecular network that is mechanically strong [3]. The physical resistance tests follow standards such as ANSI-ISEA 105-2016 [4].

Metal nanoparticle technology has been widely developed for application in several areas, such as anticancer or antibacterial applications, catalysis, photoelectric phenomena, bioengineering [5], and medical imaging, based on the special physical and chemical properties of the nanoparticles. Normally, the general synthesis methods of MNPs involve toxic chemicals that are used as stabilising and reducing agents, but this process is complex to operate and excessively environmentally unfriendly. The plant-extract-mediated biosynthesis of NMPs can be provide a number of advantages, such as ease of operation, low cost, sustainability, and environmental friendliness. Additionally, the synthesised MNPs are nontoxic, and the process produces nanoparticles with better uniformity of size than is the case with the traditional method [6]. The silver (Ag) nanoparticles have served as bactericidal agents and plasmonic nanoparticles in environmental applications. For example, Ag nanoparticles can generate reactive oxygen species (ROS), offering specific reaction selectivities and surface plasmon resonance (SPR). Actually, Ag nanoparticles have antimicrobial capabilities [7]. Ag nanoparticles had significantly higher antimicrobial activities against *Bacillus* sp. and *Escherichia coli* when compared to AgNO_3_. Furthermore, a toxicity evaluation of these Ag nanoparticles in suspension was carried out on seeds of chickpeas (*Cicer arietinum*) and mung beans (*Vigna radiata*) [8]. The zinc oxide (ZnO) nanoparticles have been reported to offer high antibacterial potential in the context of both Gram-negative and Gram-positive bacteria. The antibacterial effect of ZnO nanoparticles was demonstrated against *Staphylococcus aureus* and *E. coli* [9]. Recently, surface modifications have been applied in antibacterially coated gloves, which are a useful alternative for protecting health and public care workers from the spread of pathogenic microorganisms. The antimicrobial gloves can have advanced features in inhibiting growth or killing off microorganisms once they touch the glove surface [10].

This investigation studied the mangosteen-peel-extract-mediated biosynthesis of Ag–ZnO nanocomposites for application to antimicrobial rubber-coated gloves. The rubber gloves were tested for physical resistance under ANSI-ISEA 105-2016 standard as regards abrasion, cutting, tear, and puncture resistance, while the preparation of the liquid-natural-rubber-compound-dipped cotton gloves used a spray technique that provided a coating of Ag, Zn, or Ag+Zn nanoparticles. Subsequent antibacterial testing utilised challenges with *Pseudomonas aeruginosa, Klebsiella pneumonia, E. coli, Enterococcus faecalis,* Methicillin-resistant *S. aureus* (MRSA), and *S. aureus.*

## 2. Experimental

The chemical materials for the extract-mediated biosynthesis process were purchased as follows: 99.9% silver nitrate (AgNO_3_) from Fisher Chemical (Loughborough, UK), 98% zinc nitrate hexahydrate (N_2_O_6_Zn_6_H_2_O) from Loba Chemie PVT. Ltd. (Mumbai, India), and 89% sodium hydroxide pellets (NaOH) from Loba Chemie PVT. Ltd. (Mumbai, India). The mangosteen peel was collected from Phatthalung province, Thailand. The selected mangosteen peel was only black in colour since this is suitable for extract-mediated biosynthesis.

Other chemical materials and composites included natural latex rubber from a local factory in Surat Thani province, Thailand; 85% potassium hydroxide pellets (KOH) from Loba Chemie PVT. Ltd. (Mumbai, India); 98% potassium laurate (C_12_H_23_KO_2_) from BioFuran (Pittsburgh, PA, USA); 99.8% sulphur (S) from Loba Chemie PVT. Ltd. Mumbai, India); 97% zinc diethyldithiocarbamate (ZDC) from Sigma-Aldrich (Darmstadt, Germany); Wingstay ^®^ L from Shuguang Chemical General Co., Ltd. (Anqing, China); 99% calcium carbonate (CaCO_3_) from Sigma-Aldrich (Germany); 99% zinc oxide (ZnO) from Sigma-Aldrich (Germany); and cotton gloves from PPE Mate Company Ltd. (Don Ki Di, Thailand).

For physical resistance testing, the abrasion experiments used SODEMAT, AG04-Martindale, originally from France. The cut resistance experiments were run with SODEMAT, Coup Test, originally from France. The tear resistance and puncture resistance experiments were run with SHIMADZU, AG-X Plus, originally from Japan. All the samples were tested on the rubber-coated side of gloves (palm side).

For mangosteen-peel-extract-mediated Ag nanoparticle surface modification and its application for antibacterial purposes, the biosynthesis involved 30 g of mangosteen mixed with deionised water and boiled for 20 min at 80 °C. This mixture was then filtered with Whatman no. 1 (11 μm pore size) filter paper, and the filtrate was allowed to cool. Next, a 1-millimolar (mM) solution of silver nitrate was prepared in deionised water and mixed with 40 mL of mangosteen peel extract, while 0.1 M solution of sodium hydroxide (NaOH) was added to maintain pH 6 (indicated by the presence of dark reddish-brown coloured Ag nanoparticles [11] as the effect of pH; the organic material extracts were obtained using the silver nitrate solution preparation [12]), followed by vigorous stirring until the colour changed to dark brown, using a magnetic stirrer bar on a hot plate at 100 °C. The resulting product was a fine Ag nanocomposite prepared using mangosteen peel (Ag_mgt_).

For mangosteen-peel-extract-mediated ZnO nanoparticle surface modification and its application for antibacterial purposes, the biosynthesis involved 30 g of mangosteen mixed with deionised water and boiled for 20 min at 80 °C. This mixture was then filtered with Whatman no. 1 (11 μm pore size) filter paper, and the filtrate was allowed to cool. Further, 2 g of zinc nitrate hexahydrate solution was prepared in 100 mL of deionised water and mixed with 100 mL of mangosteen peel extract, while 0.1 M solution of sodium hydroxide (NaOH) was added to maintain pH 8 (the pH was maintained at 7.4–8 during the analysis; cotton webs were used for the seeding of isolated organism agar plates that depend on the organic material extracts obtained using the silver nitrate solution [13,14]), followed by vigorous stirring until the colour changed to white using a magnetic stirrer bar on a hot plate at 60 °C. The resulting product was a fine Zn nanocomposite prepared using mangosteen peel (Zn_mgt_).

The Ag+Zn_mgt_ nanocomposite was prepared by mixing 50% Ag_mgt_ and 50% Zn_mgt_ (mass %) and then ball milling for 12 h. The resulting product was a fine Ag + Zn nanocomposite prepared with mangosteen peel extract (Ag+Zn_msg_).

As regards the spraying technique for coating the gloves, Ag_mgt_, Zn_mgt_, and Ag+Zn_mgt_ nanocomposites were first sieved using a 100 μm sieve. The mixer combined 80 mL ethanol and 1.813 g of Ag_mgt_, Zn_mgt_, or Ag+Zn_mgt_ nanocomposite to spray on the rubber gloves. The air pressure pump was set to 7 bars, and the nozzle was located at a 12 cm distance from the gloves, with a 10 cm radius.

The liquid rubber compound and chemicals were prepared as shown in Table 1. Natural latex rubber (2000 g) was placed in the container for 1–2 min, to allow for ammonium to evaporate. After that, the 60% natural latex was poured to form a mixture with 10% KOH, 20% C_12_H_23_KO_2_, 50% S, 50% ZDC, 40% Wingstay@L, 50% ZnO, and 1% distilled water. This mixture was continuously stirred (60 rpm) as slurry for 48 h. The cotton gloves were inserted onto a solid mould for controlling the finger and palm shape. They were then dipped into the liquid rubber compound that covered the palm for 10 s, and then, the rubber gloves were sprayed with Ag_mgt_, Zn_mgt_, and Ag+Zn_mgt_ nanocomposites as shown in Figure 1, before all gloves samples were baked in the oven at 70 °C for 3 h. The analysis of crystallisation, quantity (wt%), and structure of the samples used scanning electron microscopy (SEM, EDX, Quanta 400, Prague, Czech Republic), with XRD (PANalytical Empyrean, Almelo, The Netherlands).

Concerning the challenge organisms, six isolates of bacteria were used for bactericidal activity testing of the mangosteen extracts, which were common human pathogens and drug-resistant bacteria, namely *S. aureus, K. pneumoniae, E. coli, P. aeruginosa, E. faecalis*, and methicillin-resistant *S. aureus* (MRSA). The bacterial culture was grown on nutrient agar and incubated at 35 °C for 24 h, then 2–3 single colonies were selected and kept as stock cultures with nutrient broth containing 20% glycerol at −20 °C until required for testing.

In the antimicrobial susceptibility testing, the antifungal activity of the mangosteen extracts was tested using the broth microdilution standard method following the Clinical and Laboratory Standards Institute guidelines (CLSI) outlined in document CLSI M100 [15]. Briefly, the stock suspensions of mangosteen-extract-coated Zn and Ag nanoparticles were prepared by dissolving in dimethyl sulfoxide (DMSO) and performing serial dilutions to final concentrations in the range from 5120 µg/mL to 10 µg/mL, prepared in a sterile cation-adjusted Mueller Hinton broth (CAMHB). The inoculum preparation was performed following CLSI M100 to achieve the final inoculum with 5 × 10^6^ CFU/mL.

Antimicrobial activity testing was performed using a microtiter plate. One hundred microliters of each dilution of mangosteen extract were added into wells 1 to 10. The drug-free growth control and sterility control were added to wells 11 and 12, respectively. One hundred microliters of prepared inoculum were added into wells 1 to 11. The plate was incubated at 37 °C for 24 h. The tests were performed in duplicate. Resazurin was used as a growth indicator [16]. The minimal inhibitory concentration (MIC) of each mangosteen extract was determined.

For the bactericidal activity testing of the glove surfaces, samples of nanoparticle coated gloves were prepared by cutting the coated area of the gloves into rectangular pieces of size 2 × 2.5 cm, for 5 cm^2^ area. Uncoated gloves were used as control cases. Pieces of the test and control samples were sterilised by autoclaving before testing.

Bactericidal activity testing was carried out following the ASTM D7907-14 (2019) standard method [17]. ASTM D7907 is a standard for determining whether medical examination gloves kill bacteria effectively on their surfaces. Briefly, bacterial suspensions containing approximately 10^8^ CFU/mL were prepared in sterile phosphate buffer without any interfering substances. A 10 μL bacterial suspension was distributed over the surface of the coated or uncoated glove samples and covered with a cover slip to allow contact with the glove’s surface for 0, 10, 20, or 30 min. Thereafter, the remaining bacteria on the glove’s surface were extracted with 10 mL of sterile phosphate buffer with 0.1% Tween 20. Serial dilutions were prepared and plated on tryptic soy agar (TSA) plates (BD Difco, MD, New York, USA) and incubated for 24 h at 35 ± 2 °C. The tests were conducted in triplicate, and the average bacterial colonies were calculated.

## 3. Results and Discussion

Figure 2 shows SEM image samples of Ag_mgt_ and Zn_mgt_ nanocomposites. Figure 2A shows the Ag_mgt_ nanocomposite that was interspersed on the fine carbon substrate and may be generated using mangosteen peel. The fine carbon substrate was of approximately 0.5–1.0 μm particle size. Figure 2B shows the Ag_mgt_ nanocomposite at 200,000× magnification for measuring the average 23.84 nm diameter, as presented in Table 2. Figure 2C shows the Zn_mgt_ nanocomposite revealing connections of many bubbles and high porosity, combined with the white fine carbon substrate, also generated using mangosteen peel. Figure 2D shows the Zn_mgt_ nanocomposite at 200,000× magnification for measuring the average 30.99 nm diameter, as presented in Table 2. When a nanocomposite film was composed of silver nanograins embedded in a carbonaceous matrix, created via physical vapour deposition (PVD) from separate sources of fullerene and silver acetate powders, these powders gave Ag nanograins with an average size of about 12–20 nm [18]. When an electrospray exposure system was operated at a high voltage to produce ZnO nanoparticles, they were about 20 nm in diameter [19].

Plant biosynthesis is generally related to bringing silver nitrate (AgNO_3_) into contact with the biomass/extract of plants. The manifestation of a brownish-yellow colour shade after a short duration of contact indicates the formation of Ag nanocomposite in line with the following chemical reaction [20]:Ag^+^ NO_3_^−^ + Plant molecule (OH, C=H, etc.) → Ag^o^ Nanocomposites(1)

This equation describes Ag nanocomposite synthesis conducted using plants, as reported by Shankar et al. [21]. Meanwhile, zinc oxide (ZnO) nanocomposite can act as an attractive semiconductor particle for nano-electronic and photonic applications. Highly stable and spherical 25–40 nm ZnO nanoparticles [22] had particle size that could be regulated by varying the concentration of the organic material solution [23].

Table 2 allows for a comparison of diameters between Ag_mgt_ and Zn_mgt_ nanocomposites. Notably, the Zn_mgt_ nanocomposite had ZnO as indicated by XRD in Figure 3. Based on the SEM images, diameters of Ag_mgt_ and Zn_mgt_ nanocomposites were randomly sampled as 21.85, 16.75, 27.00, and 25.85 nm and as 29.80, 23.09, 27.28, 38.93, and 35.84 nm, respectively. Hence, the average diameters of the Ag_mgt_ and Zn_mgt_ nanocomposite particles were around 23.84 ± 4.08 nm and 30.99 ± 5.73 nm, respectively. When Ag nanoparticles were formed by the reaction of 1 mM silver nitrate and 5% leaf extract of banana, Neem (Indian plants), or Tulsi leaves (Indian plants), they could be fabricated in sizes of 50 nm, 20 nm, and 50 nm, respectively [8]. Therefore, the extract-mediated biosynthesis creates particles whose diameter depends on the biomaterials. Meanwhile, the biosynthesis of *Ocimum basilicum*-leaf-extract-mediated ZnO nanoparticles has produced sizes of 20.07 ± 0.34 nm [24].

Figure 3 shows the XRD patterns of Ag_mgt_ and Zn_mgt_ nanocomposites. Figure 3A for Ag_mgt_ nanocomposite revealed a highly crystalline peak of Ag with the serial number in the XRD standard being 03-065-2871, AgCl with the number 00-031-1238, NiO with the number 03-065-6920, and C with the number 01-075-2078. Figure 3B for Zn_mgt_ nanocomposite indicated a highly crystalline peak of ZnO with the serial number in XRD standard being 01-078-3315, Zr Zn with the number 03-065-5458, and C with the number 01-075-2078.

The peaks exhibited for metallic Ag revealed five specific diffraction peaks at 2θ = 38.2, 44.4, 64.5, 77.5, and 81.6°, which indicated the (111), (200), (220), (311), and (222) reflections of metallic Ag [25]. The peaks of XRD have indicated the C element from mangosteen-peel-derived porous carbon when activated by a temperature of 800 °C [26]. The NiO and Cl elements may have been affected by the mangosteen peel to be highly biosorptive in removal of Ni [27]. The elemental concentration of mangosteen peel extract analysed with X-ray fluorescence (in wt%) has revealed 13.45% Cl, so the black-coloured peel has a high Cl fraction [28]. Regarding the ZnO, the diffraction peaks at 31.84°, 34.52°, 36.33°, 47.63°, 56.71°, 62.96°, 68.13°, and 69.18° degrees were assigned to hexagonal wurtzite phases of ZnO [29]. Generally, the mangosteen peel has obviously contained Zn, Zr, and C elements, and the black-coloured peel has high amounts of Zn, Zr, and C [28]. Therefore, the Zn elements were sourced from the mangosteen peel and the extract-mediated biosynthesis process.

Figure 4 shows SEM image samples of the surfaces of rubber gloves, namely Ag_mgt_-coated glove, Zn_mgt_-coated glove, and Ag+Zn_mgt-_coated glove. Figure 4A,B show the SEM images of the surface of the rubber glove at 200× and 1000× magnifications, for the control sample without coating. Figure 4C,D show the SEM images of the surface of the Ag_mgt_-coated rubber glove at 200× and 1000× magnifications, whereby the Ag_mgt_ nanoparticles were spherical in agglomerates, deposited on the surface of the rubber and immersed in the rubber. They were deposited at a low density on the rubber surface. Figure 4E,F show the SEM images of the surface of the Zn_mgt_-coated rubber glove at 200× and 1000× magnifications, whereby the Zn_mgt_ nanoparticles were elongated with a high aspect ratio in agglomerates, deposited on the surface of the rubber and immersed in the rubber. The deposition was of a higher density than of the Ag_mgt_ nanoparticles. Figure 4G,H show the SEM images of the surface of the Ag+Zn_mgt_-coated rubber glove at 200× and 1000× magnifications, whereby the Ag+Zn_mgt_ nanoparticles aggregates were of various shapes and also deposited on the surface of the rubber and immersed in the rubber due to the fact that when using the spray technique, the rubber still had a soft texture. This deposition was of a very high density compared to Ag_mgt_ or Zn_mgt_ nanoparticles on the rubber surfaces. The electrospinning generated 8.9–9.89 × 10^4^ g/mol ultrafine fibres with diameters of 70–175 nm and the particles were coated on the nitrile gloves [30]. Antimicrobial agents in the form of chemicals, whether naturally or chemically obtained from microorganisms, can be amalgamated into synthetic rubber (SR) or natural rubber (NR) films by coating or dispersion techniques [10].

Table 3 summarises the outcomes from physical testing of rubber gloves under the ANSI-ISEA 105-2016 standard to evaluate resistance to abrasion, cuts, tears, and punctures. Testing for abrasion resistance achieved level 1 at a load of 500 g for 100 cycles. Testing for cut resistance achieved level 3 at 1000 g, while testing for tear resistance achieved level 3 at 50 Newtons. Finally, testing for puncture resistance achieved level 4 at 100–149 Newtons. A spraying technique was used to apply coatings with Ag+Zn_mgt_, Ag_mgt_, or Zn_mgt_ on the rubber gloves, as shown in Figure 5. Figure 5A shows the Ag+Zn_mgt_ combined with carbon deposited on the rubber glove with Ag+Zn_mgt_ sharply formed, which impaled easily into the rubber glove texture. Figure 5B shows the Ag_mgt_ combined with carbon deposited on the rubber glove with Ag_mgt_ that stuck on the surface of the rubber glove. Figure 5C shows Zn_mgt_ combined with carbon deposited on the rubber glove, where Zn_mgt_ also stuck on the surface of the rubber glove. Regarding the Ag+Zn_mgt_ shape, it may be shaped in the ball milling process for 12 h. 

The main considerations when measuring the strength of the rubber gloves were the resistance properties against abrasion, cuts, tears, and punctures, measured for each of the different rubber composite textures seen in Table 1. When the resistance to abrasion was compared among NR, NR/OC/MNR (OC = organoclay; MNR = maleic anhydride grafted natural rubber), the composite using only NR had a relatively poor resistance. When eight parts of MNR were introduced, this increased the filler–rubber interactions, preventing nano layer aggregation. Moreover, the use of OC improved the rubber layer dispersion, leading to a 46% reduction in NR chains peeling from the sample surface in comparison to pure NR [31]. The findings for resistance to cuts have shown that a mixed calcium carbonate (CaCO_3_) composite exhibits greater hardness and Young’s modulus [32], leading to increased resistance to cuts. The observed resistance to tears might be attributable to an elevated crosslink concentration resulting from additional sulphur crosslinks along with the stronger modulus due to the reinforcing filler. Meanwhile, the sulphur-vulcanised NR films contain polysulphidic linkages, whereas in the peroxide-vulcanised samples, they are predominantly C to C type links (the bond strengths have rank order C-C > C-S-C > C-S-S-C) [33]. Notably, in this current study, the best resistance to punctures was exhibited by the Ag+Zn_mgt_ and Zn_mgt_ composite, where the ZnO was embedded in the rubber, thus enhancing both suspension and dispersion. It was apparent that the number of activated rubber sites increased, leading to mixing sulphur consumption in the partial interparticle crosslinking of the rubber. The production of Zn–cellulose complexes along with the elastomeric matrix might serve to support zinc dispersion in the rubber, allowing for greater resistance to punctures in the composite [34,35]. Furthermore, ZDC composite coated with nano Zn_mgt_ was used in this current study, ensuring Zn–cellulose double formation on the rubber glove surfaces.

**Table 3 polymers-17-00032-t003:** Physical test results for the rubber gloves.

Physical Test	Result	Index	Reference
Abrasion	Level 1 (ANSI-ISEA 105-2016)	- Gram load (500) - ≥100 cycles	[36]
Cut	Level 3 (ANSI-ISEA 105-2016)	- ≥ 1000 g	[36]
Tear	Level 3 (ANSI-ISEA 105-2016)	- ≥50 Newton	[37]
Puncture	Level 4 (ANSI-ISEA 105-2016)	- 100–149 Newton	[36]

The minimal inhibitory concentration of mangosteen extracts and antibacterial activities of with Ag+Zn_mgt_, Zn_mgt_, and Ag_mgt_ nanoparticles are presented in Table 4 and Figure 6. All bacterial tested isolates were susceptible to extract with Ag+Zn_mgt_, Zn_mgt_, and Ag_mgt_ nanoparticles, except for *E. faecalis*, which was not susceptible to extract with ZnO. This finding is similar to a study by Zhu et al. (2021), who reported that mangosteen extract exhibited bactericidal activity, effectively inhibiting Gram-positive bacteria, such as *S. aureus* and *Micrococcus* [38]. The potential of *α*-mangostin (AMG) derived from mangosteen extract may play an important role in antibacterial activity [39].

The MICs of mangosteen extracts with Ag+Zn_mgt_, Zn_mgt_, and Ag_mgt_ nanoparticles to susceptible bacteria ranged from 80 to 160, 40 to 320, and 40 to 160 µg/mL, respectively. All types of the mangosteen extract present bacterial inhibitory activity against both Gram-positive cocci and Gram-negative bacilli with MIC ranging in 40–320 µg/mL and against drug resistant bacteria (MRSA), with a similar MIC of 160 µg/mL. This is also corroborated by another study by Song et al. (2021), who assessed the combination of antibiotics and AMG on multidrug-resistant (MDR) bacteria. They suggested that AMG compound is promising candidate against MRSA, Vacomycin-resistant *Enterococcus* (VRE) and MDR pathogens by targeting bacterial inner membrane [30]. In our current study, we found that MICs of mangosteen extracts with metallic nanoparticles were higher than those reported earlier for a combination of antibiotics and AMG [40], and this finding could be explained by the effect of solvents on AMG yields in the extracts, for Soxhlet and ethanol solvent would provide better levels of *α*-mangostin compound than water [41]. Therefore, improving mangosteen extraction process may enhance antimicrobial efficacy of the coated gloves.

For bacterial inhibition activity of the coated glove surfaces, it was found that after 30 min contact time, the numbers of *K. pneumoniae* and MRSA were significantly reduced when tested on gloves coated with Zn_mgt_ and Ag+Zn_mgt_ nanoparticles, respectively, compared to the uncoated control sample. However, the bactericidal effect was mostly not different between the control glove and the gloves sprayed with Ag+Zn_mgt_, Zn_mgt_, and Ag_mgt_ nanoparticles, as shown in Figure 7. Based on these results, the use of antibacterial gloves may be applicable to prevent or reducing cross-contamination and indirect transmission of bacterial pathogens in general use.

## 4. Conclusions

Mangosteen-peel-extract-mediated biosynthesis of Ag_mgt_ and Zn_mgt_ nanocomposites for antimicrobial rubber-coated gloves generated particles with diameters of 23.84 ± 4.08 nm and 30.99 ± 5.73 nm, respectively. The elements Zn, Zr, and C were found in the nanocomposites. The Ag+Zn_mgt_ nanocomposite was shown to be the most densely deposited one on the surface of rubber gloves.

The rubber-coated cotton glove showed powerful, level 4 puncture resistance under ANSI-ISEA 105-2016, which is caused by Zn–cellulose complex generation supporting the dispersion of zinc in the rubber, which contained ZDC composite and was coated with nano Zn_mgt_ using a double formation on the surface of the rubber glove, thus generating stronger puncture resistance.

From the antimicrobial susceptibility testing, we found that all types of extracts effectively inhibited all 5 species of human pathogenic bacteria and one strain of important drug-resistant bacteria (MRSA). In particular, the extract with Ag_mgt_ nanoparticles presented the lowest concentration against the growth of the tested bacteria, when compared to extracts with Zn_mgt_ or Ag+Zn_mgt_ nanoparticles. All types of the mangosteen extracts presented bacterial inhibitory activity against both Gram-positive *cocci* and Gram-negative *bacilli*, with MIC ranges in 40–320 µg/mL, and against drug resistant bacteria (MRSA), with a similar MIC of 160 µg/mL.

However, when the nanoparticles were spray-coated on gloves, we found that the effectiveness in inhibiting bacteria contacting gloves surfaces had decreased. After a 30 min contact time, inhibition of *K. pneumoniae* and MRSA was found when using gloves coated with Zn_mgt_ and Ag+Zn_mgt_, respectively. It seems that the extracts actively inhibited bacteria in suspension form, while as coatings on gloves their nanoparticles needed a long incubation period of at least 30 min.

## Figures and Tables

**Figure 1 polymers-17-00032-f001:**
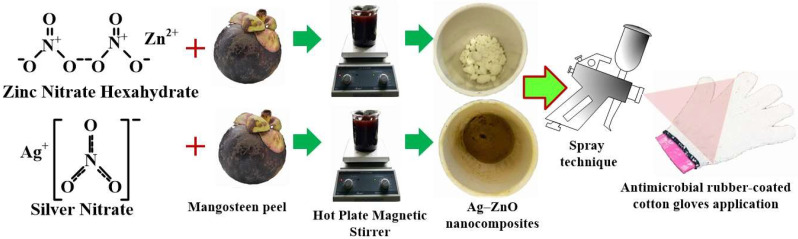
The Ag_mgt_, Zn_mgt_, and Ag+Zn_mgt_ nanocomposites were prepared and sprayed onto rubber gloves.

**Figure 2 polymers-17-00032-f002:**
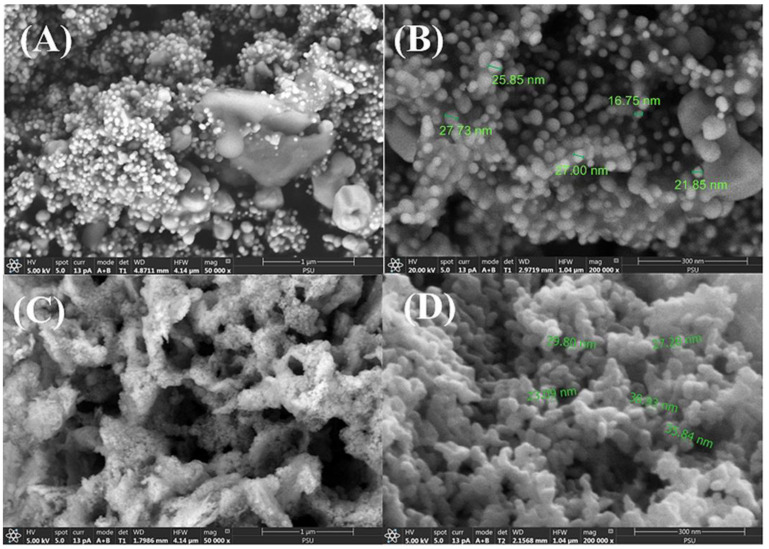
SEM image samples of Ag_mgt_ and Zn_mgt_ nanocomposites. (**A**,**B**) show the SEM images for Ag_mgt_ at 50,000× and 200,000× magnifications. (**C**,**D**) show the SEM images of Zn_mgt_ at 50,000× and 200,000× magnifications.

**Figure 3 polymers-17-00032-f003:**
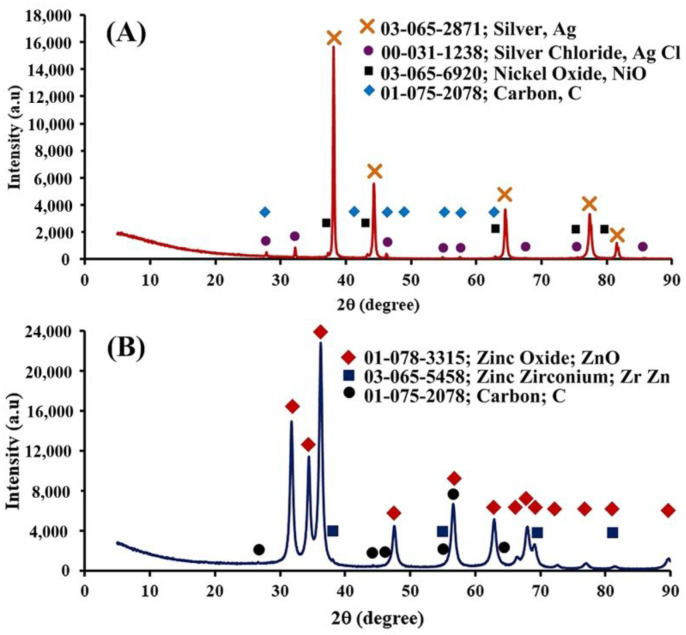
The XRD patterns of Ag_mgt_ and Zn_mgt_ nanocomposites: (**A**) the crystalline peaks of Ag_mgt_ nanocomposite and (**B**) the crystalline peaks of Zn_mgt_ nanocomposite.

**Figure 4 polymers-17-00032-f004:**
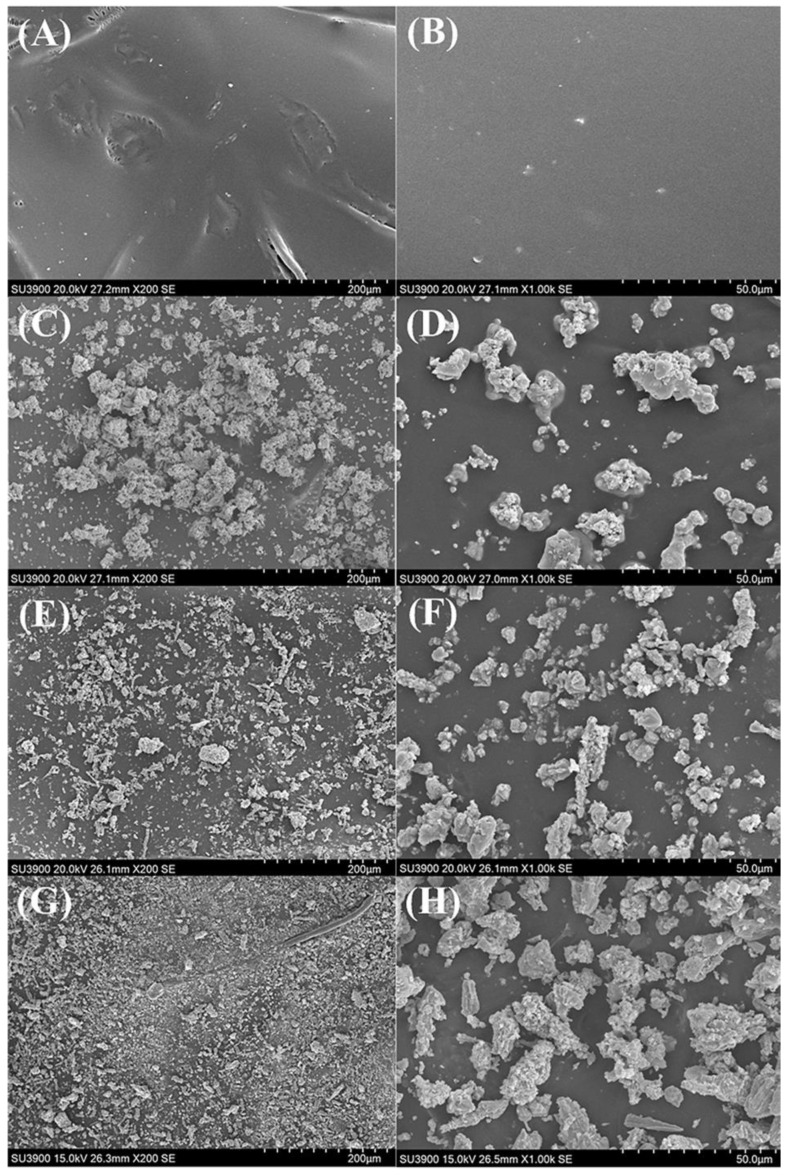
SEM image samples of the surfaces of rubber gloves; Ag_mgt_-coated rubber glove, Zn_mgt_-coated rubber glove, and Ag+Zn_mgt_-coated rubber glove. (**A**,**B**) show the SEM images of the surface of the rubber glove at 200× and 1000× magnifications. (**C**,**D**) show the SEM images of the surface of the Ag_mgt_-coated rubber glove at 200× and 1000× magnifications. (**E**,**F**) show the SEM image of the surface of the Zn_mgt_-coated rubber glove at 200× and 1000× magnifications. (**G**,**H**) show the SEM image of the surface of the Ag+Zn_mgt_-coated rubber glove at 200× and 1000× magnifications.

**Figure 5 polymers-17-00032-f005:**
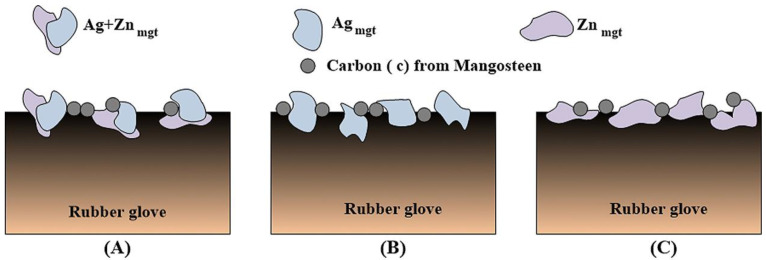
Schematic cross-sections of rubber gloves when sprayed with Ag+Zn_mgt_, Ag_mgt_, and Zn_mgt,_ to coat the surface of rubber gloves related to Figure 3 and Figure 4. The XRD patterns indicated the Ag_mgt_ and Zn_mgt_ nanocomposites, while SEM exhibited the surfaces of the Ag_mgt_ coated rubber glove, Zn_mgt_ coated rubber glove, and Ag+Zn_mgt_-coated rubber glove.

**Figure 6 polymers-17-00032-f006:**
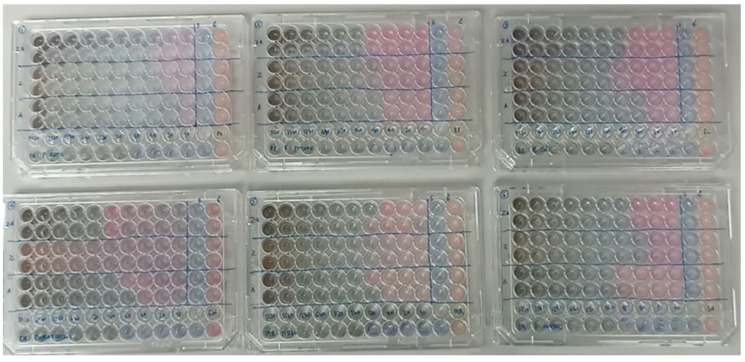
Antimicrobial activities at different concentrations of Ag+Zn_mgt_, Zn_mgt_, and Ag_mgt_ nanoparticles.

**Figure 7 polymers-17-00032-f007:**
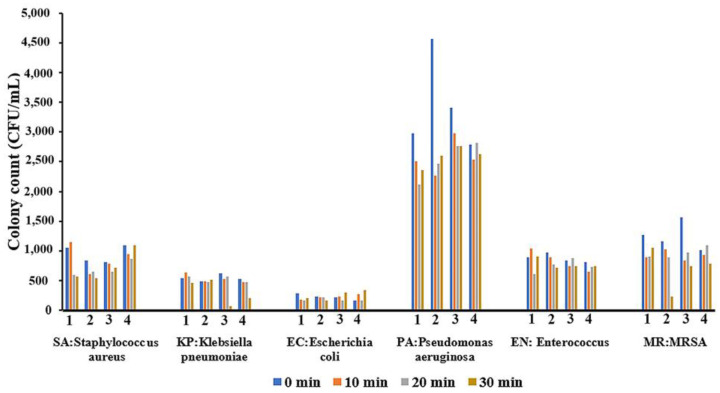
A comparison of (1) the bactericidal activity of the untreated glove, (2) Ag+Zn_mgt_, (3) Zn_mgt_, and (4) Ag_mgt_ spray coated on gloves, at contact times of 0, 10, 20, and 30 min.

**Table 1 polymers-17-00032-t001:** The ratios of materials and chemicals which were mixed for production of the antimicrobial rubber-coated gloves.

Material and Chemical Concentrations	Typical Rubber Compound Proportions by Weight (phr)
60% Natural rubber latex (NR)	100.00
10% KOH	0.30
20% Potassium laurate	0.20
50% Sulphur	0.50
50% ZDC	0.75
40% Wingstay@L	0.75
50% CaCO_3_	5.10
50% ZnO	0.40

**Table 2 polymers-17-00032-t002:** The particle diameters in Ag _mgt_ and Zn _mgt_ nanocomposites.

Nanocomposites	Diameter (nm)
Ag_mgt_	21.85
	16.75
	27.00
	27.73
	25.85
Average	23.84 ± 4.08
Zn_mgt_ *	29.80
	23.09
	27.28
	38.93
	35.84
Average	30.99 ± 5.73

* The Zn_mgt_ produced consisted of ZnO as indicated by XRD in Figure 3.

**Table 4 polymers-17-00032-t004:** Minimum inhibitory concentrations of mangosteen peel extracts against pathogenic bacteria.

Organism	Minimum Inhibitory Concentration (MIC) of Mangosteen Extract with Nanoparticles (µg/mL)		
	Ag+Zn_mgt_	Zn_mgt_	Ag_mgt_
*Pseudomonas aeruginosa*	80	80	40
*Klebsiella pneumoniae*	160	320	80
*Escherichia coli*	80	160	40
*Enterococcus faecalis*	320	>5120	160
Methicillin-resistant *Staphylococcus aureus* (MRSA)	160	160	160
*Staphylococcus aureus*	80	40	160

## Data Availability

No data were used for the research described in the article.

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
