# Peer review of "Application of Antimicrobial Rubber-Coated Cotton Gloves for Mangosteen-Peel-Extract-Mediated Biosynthesis of Ag–ZnO Nanocomposites"

_polymers, 2024, doi:10.3390/polym17010032_

Round 1
Reviewer 1 Report
Comments and Suggestions for Authors
1. In paragraph 5 of page 3, Has the Authors studied the size of NPs at pH 11, Is there any variation in size of NPs at pH 8, what is the reason for maintaining the pH at 6 and 8 respectively?
2. In page 6, SEM images are not clear, Authors should address the measurement of the size of AgNPs and ZnONPs using HR-TEM and the lattice fringe to confirm the shape and size of the particle. Size of the particles should also be compared with the XRD pattern
3. In page 6, FT-IR and UV of the extract, ZnO NPS, AgNPs are not performed, Authors can add this characterization to the manuscript.
4. In page 8, TEM, XRD, FT-IR, elemental mapping etc should be performed and added in the manuscript otherwise the material formation cannot be concluded
5. Clarify the sentence " Ag+Zn mgt nanoparticles also deposited on the surface of the rubber and immersed in the rubber"
Comments on the Quality of English LanguagePresentation of concept may be improved through proper way of expression. Clarify the sentence " Ag+Zn mgt nanoparticles also deposited on the surface of the rubber and immersed in the rubber"
Author Response
Reviewer 1
|
Revisions |
1. In paragraph 5 of page 3, Has the Authors studied the size of NPs at pH 11, Is there any variation in size of NPs at pH 8, what is the reason for maintaining the pH at 6 and 8 respectively? |
1.(Red color) It was revived at Experimental, that explained the pH and referenced pH. |
2. In page 6, SEM images are not clear, Authors should address the measurement of the size of AgNPs and ZnONPs using HR-TEM and the lattice fringe to confirm the shape and size of the particle. Size of the particles should also be compared with the XRD pattern. |
2.(Blue color) SEM in Figure 4 was rewritten to explaining nanoparticle AgNPs and ZnONPs size and involved to Ag and ZnO particle for confirmed with peaks. For HR-TEM was not measured particle size but it can be confirmed by SEM. |
3. In page 6, FT-IR and UV of the extract, ZnO NPS, AgNPs are not performed, Authors can add this characterization to the manuscript. |
3.FT-IR, UV were not measured due to take more than 7 days. |
4. In page 8, TEM, XRD, FT-IR, elemental mapping etc should be performed and added in the manuscript otherwise the material formation cannot be concluded. |
4.TEM was not measured due to take more than 7 days. (Blue color) XRD was measured peaks confirmed the Ag and ZnO nanoparticles. |
5. Clarify the sentence " Ag+Zn mgt nanoparticles also deposited on the surface of the rubber and immersed in the rubber" Presentation of concept may be improved through proper way of expression. |
5.(Blue color) Figure 4 (G-H) was added more explanations of Ag+Zn mgt nanoparticles also deposited on the surface of the rubber and immersed in the rubber. |

Reviewer 2 Report
Comments and Suggestions for Authors
The paper is describing a treatment of gloves surface with a complex mixture whichis prepared by the participation of a natural product ,but the structure of this last is not indicated and the action mechanism is not depitched.1.The abstract is rather an anticipation of the discussion rather than a description of the work reported in the paper, The material preopare is rather NR filled with various components rather than a composite.
The part of interest for polymer science is very minor and not technically supported adequately.
Some points needing improvement are in any case indicated.
2.The keyword should include reference to the antimicrobial activity
3.Figure 2 is trivial and not necessary
4,Information about the chemical reaction involving the claimed formation of Ag and ZnO nanoparticles is completely lacking.
5.In the discussion related to the nanocomposite metal particles diameter and Table 2 the zinc is reported as Zn ; should be ZnO ??
6.Table 3 : it is not clearly indicated the substrate on which the characterization was performed .
/.The long discussion following table 3 is quite difficult to follow and based on comparison not flakedand supported by numerical values in the various cases compared. The approach should be more scientific.
Comments on the Quality of English Language
Some substantial improvemet in style would help
Author Response
Reviewer 1 |
Revisions |
1. The paper is describing a treatment of gloves surface with a complex mixture which is prepared by the participation of a natural product ,but the structure of this last is not indicated and the action mechanism is not depitched.1.The abstract is rather an anticipation of the discussion rather than a description of the work reported in the paper, The material preopare is rather NR filled with various components rather than a composite. The part of interest for polymer science is very minor and not technically supported adequately. Some points needing improvement are in any case indicated. |
1.(Violet color) The action mechanism was explained at Results and Discussion (Violet color) Abstract was rewritten to cover shortly all results. (Green color) Table 3 was more explained polymer science based on the chemical reaction that effected to physical test such as Abrasion, Cut, Tear, Puncture. |
2.The keyword should include reference to the antimicrobial activity. |
2.Keywords was added Antimicrobial activity |
3.Figure 2 is trivial and not necessary. |
3.Figure 2 (SEM) is very important that confirmed the nano size and infrastructure of Agmst and Znmst |
4. Information about the chemical reaction involving the claimed formation of Ag and ZnO nanoparticles is completely lacking. |
4.(Violet color) Figure 2 was the action mechanism was explained formation of Ag and ZnO at Results and Discussion. (Green color) Table 3 was more explained polymer science based on the chemical reaction that effected to physical test such as Abrasion, Cut, Tear, Puncture. (Blue color) Figure 4 (G-H) was added more explanations of Ag+Zn mgt nanoparticles also deposited on the surface of the rubber and immersed in the rubber.
|
5. In the discussion related to the nanocomposite metal particles diameter and Table 2 the zinc is reported as Zn ; should be ZnO ?? |
5.(Brown color) The explanation of Table 2 was added the ZnO |
6.Table 3 : it is not clearly indicated the substrate on which the characterization was performed . |
(Green color) Rewritten for easier reading, Table 3 was more explained polymer science based on the chemical reaction that effected to physical test such as Abrasion, Cut, Tear, Puncture.
|
7. The long discussion following table 3 is quite difficult to follow and based on comparison not flaked and supported by numerical values in the various cases compared. The approach should be more scientific. |
6.(Green color) Rewritten for easier reading with more scientific explanation. |

Round 2
Reviewer 2 Report
Comments and Suggestions for Authors
Thnks for the extended revision and for considering my comments.I think the paper is now substantially improved for publication apart a revision of the english usage
Comments on the Quality of English LanguageA full style revision would be very useful even if the paper is presently understandable
Round 2
Comments
Thnks for the extended revision and for considering my comments. I think the paper is now substantially improved for publication apart a revision of the English usage
Revision
The manuscript was proofed English by native speaker.
